# DNA replication timing reveals genome-wide features of transcription and fragility

Francisco Berkemeier [1,2] ✉, Peter R. Cook [3] & Michael A. Boemo [1,2] ✉

DNA replication in humans requires precise regulation to ensure accurate genome duplication and maintain genome integrity. A key indicator of this regulation is replication timing, which reflects the interplay between origin firing and fork dynamics. We present a high-resolution (1-kilobase) mathematical model that infers firing rate distributions from Repli-seq timing data across multiple cell lines, enabling a genome-wide comparison between predicted and observed replication. Notably, regions where the model and data diverge often overlap fragile sites and long genes, highlighting the influence of genomic architecture on replication dynamics. Conversely, regions of strong concordance are associated with open chromatin and active promoters, where elevated firing rates facilitate timely fork progression and reduce replication stress. In this work, we provide a valuable framework for exploring the structural interplay between replication timing, transcription, and chromatin organisation, offering insights into the mechanisms underlying replication stress and its implications for genome stability and disease.

Accurate DNA replication is essential for faithfully duplicating genetic information, ensuring its preservation for future generations[1]. In humans, replication occurs during S phase when multiple discrete chromosomal sites, termed origins of replication[2], fire to initiate bidirectional replication forks−molecular machines that traverse the chromosome and replicate DNA[3]. These forks move in opposite directions, progressing until they encounter another fork, reach a chromosome end (Fig. 1a), or face an obstacle (e.g., a bound protein or transcription complex[4]). Intriguingly, each origin fires stochastically so firing sites and times differ from cell to cell. Despite this apparent randomness, consistent trends emerge so that different cell types have characteristic firing profiles[5].

Replication timing refers to the time at which a specific locus either fires (if an origin) or is passively replicated by an incoming fork. These timing profiles are closely associated with various chromatin structures[6], as well as gene expression[7] and replication stresses[8]. Furthermore, timing is linked to genetic variation[9] and cancer (where late or delayed replication often correlates with increased genomic instability[10]). Of particular interest are fragile sites, regions that are especially vulnerable to breakage due to replication stress, and are often found in late-replicating regions[11]. These sites, and the long genes

found within them, are often hotspots for the chromosomal rearrangements and deletions that arise in cancers and other genetic diseases[12].

Replication, transcription, and chromatin organisation are also intricately inter-connected, with each influencing the other[13–15]. In particular, chromatin remodelling regulates the accessibility of regulatory factors, influencing both gene expression and replication. Open chromatin is strongly linked to transcriptional activity and plays a crucial role in replication timing[16,17]. Although associations between genomic features are well-established, identifying site-specific or context-dependent differences remains a challenge. Experimental approaches often struggle to isolate individual variables, limiting our ability to disentangle the interplay between replication and other processes.

To address these gaps, we develop a stochastic model that maps origin firing rates to replication timing, capturing variability across cell populations. By integrating data from RNA-seq[18], ChIP-seq[19], GRO-seq[20], and a database of fragile sites (HumCFS[21]), we provide a framework to explore how discrepancies between the model's predictions and experimental data may reflect signatures of transcriptional activity, chromatin openness, and genomic fragility. Our model acts as a

[1]Department of Pathology, University of Cambridge, Cambridge, UK. [2]Department of Genetics, University of Cambridge, Cambridge, UK. [3]Sir William Dunn School of Pathology, University of Oxford, Oxford, UK. ✉e-mail: fp409@cam.ac.uk; mb915@cam.ac.uk

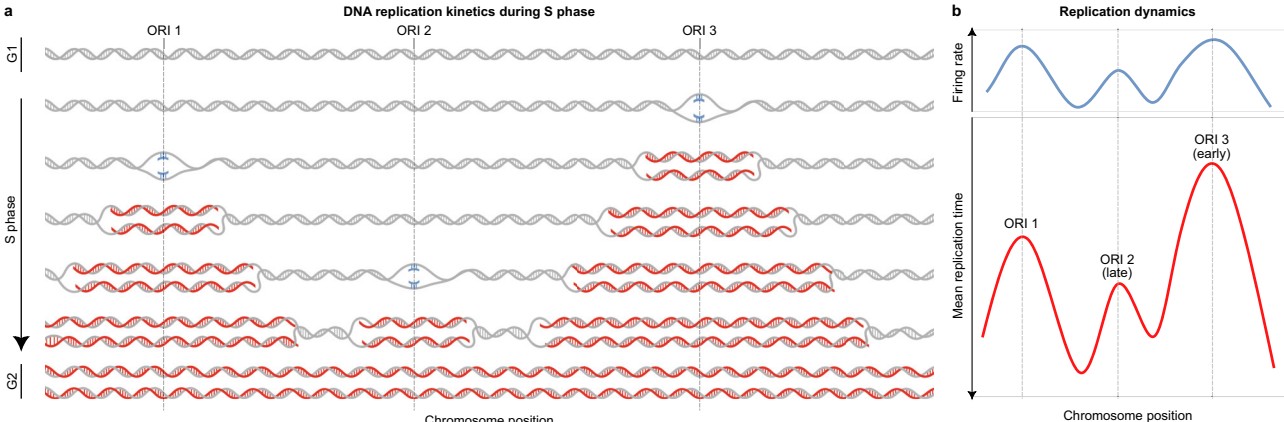

**Fig. 1 | A kinetic model of DNA replication. a** Replication initiates at specific origins that are licensed by the end of G1 phase. During S phase, replication forks progress bidirectionally from origins, passively replicating DNA until they merge with forks from adjacent origins or reach chromosome ends to complete replication and enter G2. In this example, three origins (ORIs 1, 2, and 3) fire at different times, with nascent DNA strands shown in red. At the end of replication, two identical copies of the original template are formed. **b** Illustration of the expected inverse but non-trivial correlation between firing rates (top) and replication timing (bottom, with an inverted y-axis). In a model where the firing time of each origin is an exponentially distributed random variable, the firing rate is the parameter of this distribution and tends to decrease as replication timing increases, indicating that regions with higher firing rates replicate earlier in S phase. Replication timing, measured by Repli-seq, shows the average replication time across a cell population, with peaks corresponding to potential origins. ORI 2 is in a late-replicating region, while ORI 3 replicates earlier, as indicated by their relative positions on the timing curve. Adapted from Hulke et al.[70].

null hypothesis, representing how replication should occur in the absence of perturbation from genomic features. The central aim is to identify loci where the model's predictions diverge from experimental observations, highlighting regions that may experience replication stress or other anomalies. By deriving a closed formula for the expected time of replication at each genomic site, we establish a solid mathematical framework to support our computational simulations.

Our workflow is simple: using only timing data as input, along with minimal genomic parameters such as potential origin locations, the model determines firing rates and predicts timing profiles plus other key kinetic features like fork directionality and inter-origin distances. Researchers with replication timing data can use this model to rapidly generate precise replication dynamics profiles without extensive computational expertise, revealing factors that influence replication timing and genome instability across various contexts.

Despite significant advances in mathematical modelling[22–25], deriving a position-specific, data-fitted model that precisely links replication timing to origin firing has remained a challenge. While some approaches rely on neural networks to infer probabilistic landscapes of origin efficiency[26], ours differs by deriving a closed-form relationship between timing and firing. Rather than relying on complex inference techniques, our model abstracts intrinsic firing rates without directly tying them to specific biological mechanisms such as licensing or activation. This allows a precise fit to observed timing data and enables simulation of genome-wide dynamics in a direct and interpretable manner. Our approach improves existing fitting methods by adopting a convolution-based interpretation of the timing programme. Using process algebras from concurrency theory[27], we model replication forks and origins as a concurrent system, simulating their behaviour across the genome. In this work, we demonstrate how a theoretical description of replication timing uncovers key links between timing, genomic stability, and other essential genomic processes.

## Results

### An equation for DNA replication timing
We begin by introducing our stochastic framework for replication timing, which is fully detailed in the Methods and in Supplementary Note 1. We aim to identify and quantify genomic regions where replication timing deviates from theoretical predictions, hereafter referred

to as replication timing misfits, which may indicate potential sites of replication stress or instability. To accomplish this, we model the complex, nonlinear relationship between origin firing rates and replication timing (Fig. 1b) and fit these rates to experimental timing data. This approach enables investigation using replication forks, origins, and DNA templates as the level of abstraction.

In our framework, the genome is divided into 1 kb segments (sites). Each site $j$ can fire as an origin at a rate $f_j$, while replication forks progress at a constant speed $v$. Concretely, the waiting time for each site's origin to fire follows an exponential distribution with parameter $f_j$. Let $T_j$ be the time at which site $j$ is replicated, either by firing as an origin itself or by being passively replicated by an incoming fork. The expected replication time at any site $j$ is then obtained by weighting the contributions from all potential origins, leading to the following closed-form expression

$$\mathbb{E}[T_j] = \sum_{k=0}^{R} \frac{e^{-\sum_{|i| \le k}(k-|i|)f_{j+i}/v} - e^{-\sum_{|i| \le k}(k+1-|i|)f_{j+i}/v}}{\sum_{|i| \le k} f_{j+i}} \quad (1)$$

where the indices $\{j \pm i\}$ cover neighbouring loci within a chosen radius of influence $R$, i.e., the distance within which neighbouring origins are assumed to affect the timing of a focal origin. Equation (1) enables us to infer the stochastic model's firing rates $\{f_j\}$ from timing data (e.g., Repli-seq), generating a best-fit timing profile for the entire genome that can then be compared with the observed measurements. Regions exhibiting significant discrepancies (misfits) can indicate replication stress or other biological factors not captured by the model. In the following sections, we apply this model to different human cell lines, demonstrating how it reproduces global replication patterns and highlights specific genomic loci that may warrant deeper investigation.

### Predicting genome-wide replication
After assigning the time of replication (determined using Repli-seq data) to every 1 kb segment of the genome in 11 different human cell lines, site-specific firing rates are fit to the data via Eq. (1). Then, replication is simulated using Beacon Calculus (`bcs`), a concise process algebra ideal for concurrent systems (see Supplementary Note 2). Finally, we explore patterns of replication seen after averaging 500 simulations for each of the 11 lines (Fig. 2a).

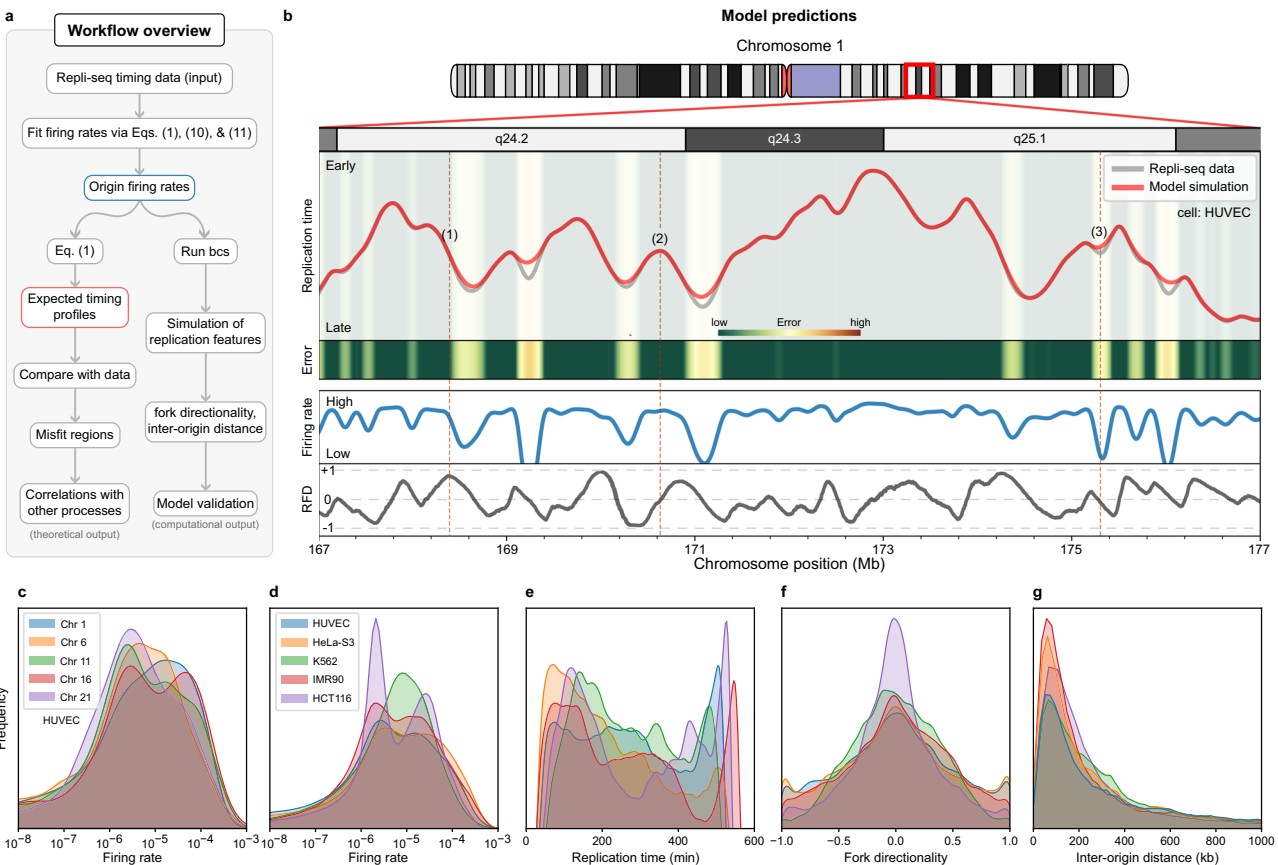

**Fig. 2 | Predicting genome-wide features of replication. a** Overview of the main model and analysis. Starting with Repli-seq timing data, origin firing rates are fitted through Eqs. (1), (10), and (11). These rates generate expected timing profiles for comparison with experimental data to identify regions of timing misfits and fork stalling, which are analysed for correlations with other genomic processes. Simulations of replication features, such as fork directionality and inter-origin distances, validate the model against the literature. **b** Example of main modelling outputs from a region in HUVECs. Here we see the replication timing of both experimental and simulated data, and the magnitude of the misfit (error) for replication timing in a region where replication forks often stall; this leads to elevated errors that the model struggles to capture accurately. We also show the inferred origin firing rates and fork directionality, scaled between -1 (leftward) and +1 (rightward). We

highlight three regions of interest: (1) A passively replicated site predominantly replicated by rightward-moving forks (RFD ≈ 1); (2) A likely origin, characterised by a high firing rate and an RFD of 0; (3) A poorly fitted region between two origins with a low firing rate determined by the fitting algorithm with RFD of 0 (an equal likelihood of replication by leftward- and rightward-moving forks). **c** Kernel density estimate (KDE) of firing rate distributions across selected chromosomes in HUVECs. **d**–**g** KDEs comparing genome-wide features—including firing rates, replication timing, fork directionality, and inter-origin distances—across different cell lines. All distributions align with experimental observations. Areas under curves are equal to 1, while y-axis values are omitted to emphasize relative shapes and distributions rather than absolute magnitudes.

We begin by comparing experimental timing profiles to those obtained from Eq. (1). Note that this is equivalent to averaging the timing profiles from a large number of `bcs` simulations, which also allows us to save significant computational resources when computing timing alone. An example for chromosome 1 in HUVECs is shown in Fig. 2b. As expected, some regions replicate early (e.g., around 173 Mb) and others late (e.g., around 171 Mb). Overall, the model's predictions agree well with the experimental data (see Methods).

We focus on the regions with high misfit error (shaded in yellow and red in Fig. 2b), where the assumption of constant fork speed in Eq. (1) leads the model to predict earlier replication times than are observed experimentally. Because a constant speed imposes an inherent upper bound on how quickly replication can transition from early to late (see Supplementary Note 2.2.2), any steeper or delayed transitions—potentially arising from fork stalling or local inefficiencies—remain unmatched by the model. These high-misfit zones thus highlight loci where forks appear to slow or stall beyond our simplified assumptions, flagging potential replication stress hotspots for more detailed study.

While firing rates are directly inferred from Eq. (1), replication fork directionality (RFD) is calculated as the proportion of cell cycles (or

`bcs` simulations) in which a given site is replicated by rightward versus leftward forks. RFD values range from −1 (always replicated by leftward forks) to +1 (always replicated by rightward forks), with intermediate values indicating a mix of replication directions across simulations (Fig. 2b).

To validate the model, we examine global distributions of multiple features. Despite little variation in firing in HUVECs (Fig. 2c), HCT116 exhibits a pronounced bimodal pattern, likely driven by differences in data sources[28] (Fig. 2d, e), which may affect how replication timing and origin firing rates are captured. Regarding RFD, our results demonstrate a balanced bidirectional fork movement, with fork directionality symmetrically distributed and accumulating around zero, indicating efficient replication progression (Fig. 2f). This pattern aligns with recent quantifications of fork directionality in human cells[29]. While determining inter-origin distances (IODs) is straightforward from our simulations, doing so from DNA-fibre experiments remains challenging due to technical limitations and potential biases[30]. Nevertheless, simulations show a concentration of IODs within the commonly observed range of 100–200 kb[31] (Fig. 2g).

Although these results validate the model against established metrics, its broader ability to simulate other features, like replicon

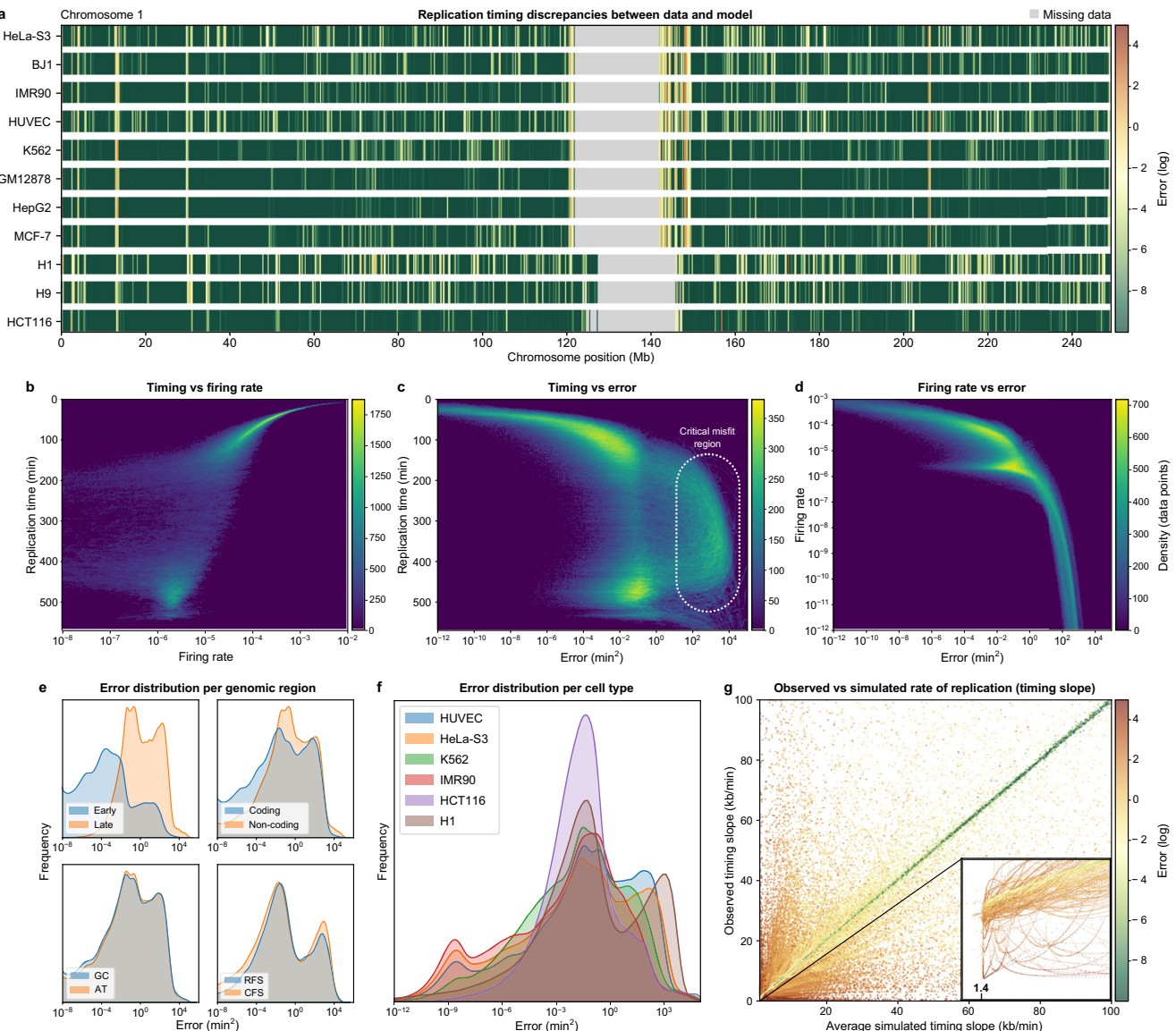

**Fig. 3 | Detecting discrepancies in replication timing determined experimentally and in simulations. a** Normalised error plots (red−high error, green−low error) highlighting deviations between simulated and experimental replication timings (chromosome 1 in various human cell lines). Grey areas: missing or unavailable data. **b−d** Density scatter plots illustrating key relationships in H1 cells (averages of 500 simulations). Pairwise combinations of three variables are shown: replication time, firing rate, and error. In **b**, the inverse correlation between replication timing and firing rate is evident, with greater variability in firing rates late in S phase. **c** shows the relationship between replication timing and error, revealing that high errors are distributed throughout S phase (dotted oval). **d** illustrates the branching relationship between firing rate and error. **e** Error distributions in HUVEC cells, grouped by replication timing (early vs. late), genic vs. intergenic regions, GC vs. AT content, and classification of fragile sites (common vs. rare, CFS vs. RFS). **f** Genome-wide error profiles in different cells. **g** Scatter plot comparing average simulated timing slope, indicative of the progress of replication over time, against observed data, colour-coded by associated error. The zoomed-in region at [1.2, 2] × [0, 2] kb/min highlights the 1.4 kb/min bound on the simulated slope. Each dot represents a simulated-observed data pair, with the strand-like continuity arising from the high resolution of our 1 kb model, where proximity between adjacent pairs reflects the minimal positional shifts captured at this scale.

lengths and active fork numbers, highlights its value in capturing the full spectrum of replication dynamics. The most compelling insight, however, comes from examining regions where the model's predictions diverge from data, as these discrepancies may coincide with critical sites of genomic instability, revealing areas of unique biological interest, which we address next.

## Hotspots of instability

We now determine genome-wide error profiles in all 11 cell types (Fig. 3a illustrates those for chromosome 1). Remarkably, some of the regions that fit poorly are found in all cell lines (despite using different genome builds); this underscores the robustness of profiles across cell types[32]. Replication timing and firing rates show a strong negative correlation (Spearman's rank correlation of ~ −0.89; Fig. 3b); regions with higher firing rates tend to replicate earlier. Late-replicating regions also have a wide spread of low firing rates, reflecting a pattern captured by the fitting algorithm. Additionally, the lowest errors are seen in the earliest replicating regions, moderate ones in both early- and late-replicating regions, and the highest are distributed throughout mid-to-late S phase (Fig. 3c). This suggests misfits increase as S phase progresses and fewer firing events occur. Low firing rates are also associated with high errors (Fig. 3d; note the branched profile, reflecting difficulties in accurately modelling high-to-low firing rate transitions). Timing misfits are

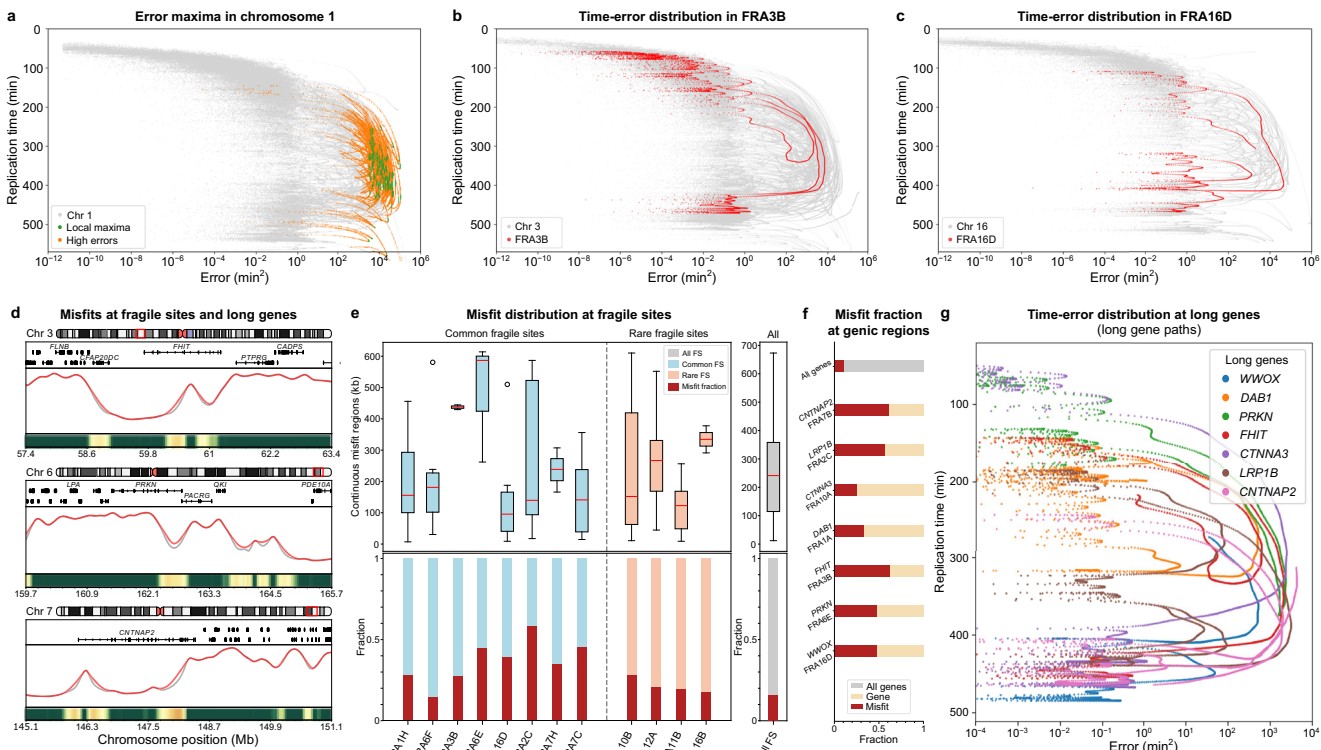

**Fig. 4 | Timing errors in fragile sites and long genes. a** Replication timing vs. error on chromosome 1 in H1, highlighting regions with local maxima in error and neighbouring high-error zones (within a 300 kb radius). The threshold for identifying local maxima in errors is set at $10^{2.8}$ (min²). Each dot represents an error-timing data pair, with the strand-like continuity arising from the high-resolution of our 1 kb model. **b, c** Genome-wide scatter plots displaying replication timing vs. error, with specific focus on common fragile sites FRA3B and FRA16D, revealing a continuous error path in mid-to-late replication, near the *FHIT* and *WWOX* genes, respectively. **d** Examples of misfit regions detected by the model across three different chromosomes (3, 6, and 7). Each panel shows the chromosome ideogram, gene locations, and a comparison between the observed data (grey) and model predictions (red), as well the associated error. Notably, misfit regions overlap with long genes such as *FHIT* (Chr 3), *PRKN* (Chr 6), and *CNTNAP2* (Chr 7). **e** Misfit distribution for common (blue) and rare (pink) fragile sites, compared with the total fragile site misfit fraction (grey). Top: length (in Mb) of continuous misfit regions. Box plots show the median (middle line), 25th–75th percentiles (box), whiskers up to 1.5 times the interquartile range, and outliers (open circles). In total, 1262 continuous misfit regions across all fragile sites were analysed to illustrate global trends. Bottom: normalised misfit fraction at different sites. **f** Misfit fraction analysis of whole-genome genic regions, and at the largest genes within fragile sites (normalised). **g** Scatter plot of replication timing vs. error trajectories for long genes, highlighting error accumulations based on gene size and location within fragile sites.

predominantly concentrated in late-replicating regions (Fig. 3e). This is consistent with prior results suggesting that the replication machinery encounters more obstacles towards the end of S phase[33,34]. Additionally, errors exceeding $10^4$ (min²) are more frequent in non-coding regions compared to coding ones, indicating a potential vulnerability of non-coding DNA to replication stress. Misfits also vary between cell lines, with HCT116 displaying a distinct pattern likely due to differences in data processing (Fig. 3f; see Methods). Similar disparities were observed in firing rate distributions, hinting at the potential for cell line-specific analyses to offer further insights. However, given our focus here, we leave a detailed analysis of these dynamics for future exploration.

In regions with infrequent origin firing, the slope of the timing curve—representing the rate of replication changes over time—is primarily governed by fork speed, establishing an effective bound of 1.4 kb/min (Fig. 3g). This constraint becomes most evident in regions where observed gradients fall beyond such a bound, resulting in error accumulation around slower-replicating areas. Origin competition, where nearby origins fire at similar times, further compounds these errors, producing timing valleys between origin firing peaks. These patterns highlight regions of potential stress, suggesting areas for further study.

## Fragile sites and long genes

Fragile sites are specific chromosomal regions prone to gap formation or breakage under conditions of replication stress[35]; examples include FRA3B[36] and FRA16D[37]. They frequently arise after inhibiting DNA synthesis or applying other replication stresses[38], often contain few origins[11], and likely result from fork stalling or collapse[39]. Fragile sites can be broadly categorized into common fragile sites (CFSs) present in most of the population and rarer ones (RFSs) found in relatively few individuals[21,40]. As seen before, replication timing misfits tend to be most pronounced in mid-to-late S phase, where regions such as fragile sites often coincide with higher errors (Fig. 4a–d). FRA3B and FRA16D show even greater median misfit lengths (Fig. 4e), indicating that these loci can pose particular challenges for our model. Likewise, large genes in fragile sites (e.g., *CNTNAP2*, *LRP1B*, *FHIT*) exhibit substantial errors (Fig. 4f, g), and applying an error threshold confirms that long genes overlapping misfit regions (~30% of which reside in fragile sites) stand out readily (Supplementary Table 1). Although certain chromosomes (e.g., 15, 20, 22) lack major fragile-site misfits, most display a notable linkage between fragile sites, large genes, and replication stress. While not all fragile sites follow one uniform pattern, these observations underscore broad genomic regulation factors that potentially influence replication timing and error.

Fragile sites are also known to be cell-type specific[36,41], yet our analysis of 11 lines, with H1 cells serving as one illustration, suggests that core fragility-misfit correlations are relatively consistent even if the degree of disruption varies. A more detailed case study of HCT116, where confirmatory data on fragile site expression is available[42], is

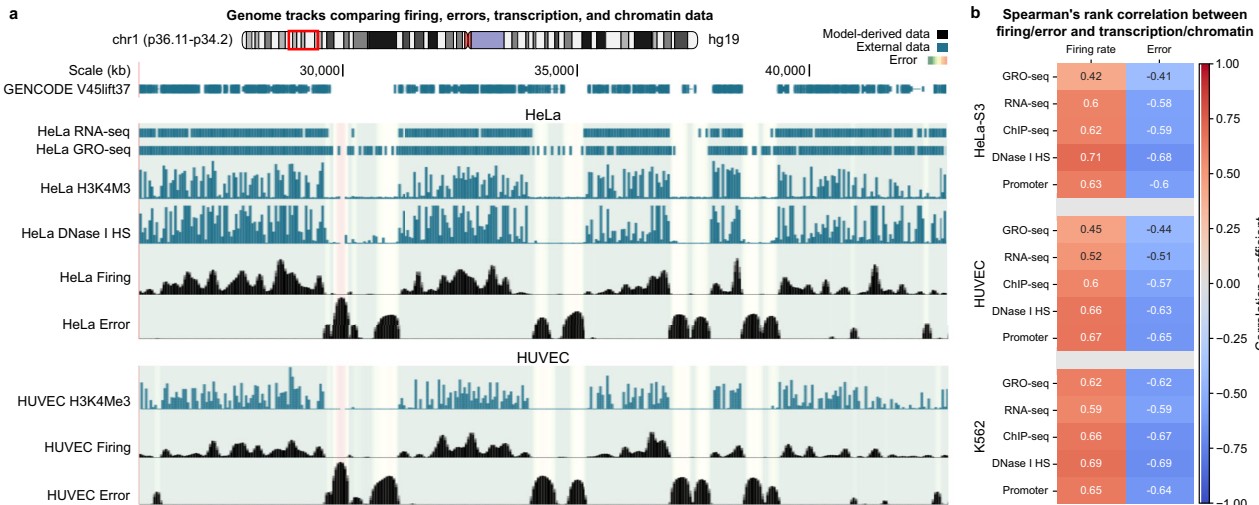

**Fig. 5 | Replication timing discrepancies and firing rate profiles correlate with transcriptional and chromatin data. a** Snapshot from the UCSC Genome Browser showing a detailed view of chromosome 1 (p36.11-p34.2) in HUVEC and HeLa (hg19). Various tracks compare transcriptional and chromatin data to misfit magnitude (error) and firing rate profiles obtained from our model (log-scale). Tracks include RNA-seq (marking mature mRNA levels), GRO-seq (nascent RNA), ChIP-seq for H3K4Me3 (promoters), and DNase I hypersensitivity (open chromatin). The error for each line is represented as a translucent heat map across tracks, with colours ranging from green (good fit) to yellow/red (poor fit). **b** Heatmap displaying the Spearman correlation coefficients between origin firing rates and fit errors with transcriptional and chromatin features for HeLa, HUVEC, and K562. All tests (two-sided) returned *p*-value < 10$^{-15}$.

included in the Supplementary Information (Supplementary Note 2.4), highlighting how individual chromosomes and cell-line dependent features can shape replication stress at fragile sites. Taken together, these findings reinforce that fragile sites often correlate strongly with replication misfits, though not uniformly across the genome or in every cell type. By pinpointing likely hotspots of replication stress, our model provides a powerful framework for guiding experimental follow-up.

## Transcription and chromatin state

Transcription and replication have long been recognised to interact in complex and sometimes conflicting ways, particularly at fragile sites[43]. Previous studies show that transcription can create barriers to replication, mainly via R-loops, that can obstruct fork progression, leading to stalling or collapse. Long genes associated with CFSs have a scarcity of replication origins, forcing forks to traverse a long distance which can delay replication[44]. This delay is particularly pronounced in transcriptionally active regions. However, this is not always the case, as chromatin structure can play a more dominant role in timing discrepancies. Building on the previous results, we now turn our attention to interactions between transcription, chromatin structure, and replication. Regulatory elements like active promoters and enhancers are marked by histone modifications such as H3K4me3[45], DNase I hypersensitivity (DHS), and transcription-factor binding, detected using ChIP-seq[46,47]. By integrating data from ChIP-seq, RNA-seq, and GRO-seq[18,20,46,48], we assess how these markers are associated with replication timing. Regions with high GRO-seq signals align with peaks in H3K4me3 and DHS signals; they exhibit lower timing errors and higher firing rates (Fig. 5a). Spearman rank correlation analyses reveal varying degrees of association between variables (Fig. 5b). This method was chosen due to its suitability for non-normally distributed data and its ability to capture monotonic relationships, reflecting the ranked nature of our genomic features. Pearson and Kendall's Tau tests were conducted for comparison (Supplementary Note 2.5). The consistently higher Spearman rank correlations indicate a strong monotonic relationship, particularly between DHS sites and firing rates, as well as between promoters and firing rates, revealing how chromatin accessibility facilitates

replication initiation, even amid non-linear interactions. We observed a moderate to strong negative correlation between GRO-seq and misfits across all lines. This suggests that the replication machinery may encounter fewer impediments in regions with active transcription. A possible explanation is that transcriptionally active regions are more likely to be in an open state, reducing mechanical barriers to fork progression and lowering the chances of replication stress. Moreover, transcription factor-binding sites have been shown to enhance DNA replication, as evidenced by studies demonstrating that these sites significantly increase replication efficiency[15].

Furthermore, origin density strongly correlates with promoter density[13]. This co-evolution of replication and transcription regulatory regions further supports the idea that transcriptional activity not only facilitates replication but also influences the efficiency and organisation of origins in mammalian cells. The strong correlation between high origin firing rates and regions of active transcription, open chromatin, and promoters provides further insight into genome-wide coordination of replication and transcription. Notably, putative origins are often located in open and early-replicating chromatin[17,49] that is well fitted by our model. This synchronisation between replication and transcription may prevent replication stress, particularly during late S phase, aligning with the observation that transcriptionally active euchromatin tends to replicate early, and silent heterochromatin late[50]. Under replication stress, this coupling is adjusted, with initiation and termination sites shifting to maintain the balance between replication and transcription, highlighting the intricate coordination that sustains genome integrity[51].

## Discussion

In genome-wide simulations, our model effectively captured key replication dynamics, including replication timing, fork directionality, and inter-origin distances. Replication timing was fitted with high precision across most of the genome, with only a few regions where observations clearly deviate from simulations. While misfit distributions varied across different chromosomes and cell lines (Fig. 2), late-replicating regions consistently exhibited higher misfit rates (Fig. 3). This matches previous findings suggesting these regions are more prone to replication challenges. Firing rates were also strongly

negatively correlated with timing misfits; regions with infrequent origin firing are more susceptible to timing deviations. Additionally, noncoding regions had a higher frequency of misfits, highlighting their potential vulnerability.

We found that many replication-timing misfits occur in proximity to fragile sites and long genes (Fig. 4). Our analysis pools data from multiple studies and cell types[21]. While this provides a broad overview, it does not account for the cell type-specific replication programmes that underlie fragile site expression. Fragile sites are influenced by transcriptional activity and replication timing, both of which vary between cell types[41,52]. For instance, fibroblasts and lymphoblastoid cells exhibit distinct replication initiation patterns, which affect the timing and extent of fragility[36]. We observed consistent trends in the correlation between fragility and replication timing misfits across all 11 lines analysed, with H1 cells used as an illustrative example. This consistency highlights the robustness of the approach in identifying conserved replication dynamics and suggesting candidate regions and genes of interest.

We also performed a statistical assessment of misfit distributions in HCT116, a cell line with robust confirmatory data on fragile site expression[42]. Our results indicate that although fragile sites in HCT116 frequently show statistically significant differences in replication misfits, especially on certain chromosomes, this pattern is not uniform across the entire genome. Further targeted studies could help clarify how cell-line-specific factors shape localized vulnerability within a broader replication error landscape. At the same time, by pinpointing potential replication stress hotspots, our model provides a valuable foundation for deeper experimental investigations into the molecular underpinnings of fragility. Researchers with access to cell type-matched Repli-seq and fragility data could refine this framework to achieve more specific insights.

Additionally, we note that early-replicating fragile sites (ERFS), often linked to highly transcribed genes[53], represent another compelling avenue for future work. For instance, the study by Tubbs et al.[54] demonstrates that poly(dA:dT) tracts can precipitate replication fork collapse in both early- and late-replicating domains, suggesting that similar sequence features may underlie fragility across diverse replication windows. However, robust, high-resolution data on ERFS in human cell lines remain limited; most existing datasets come from mouse[53] or avian cells[55], and cross-species mapping (e.g., via LiftOver) does not reliably capture species-specific replication landscapes. Once comprehensive human datasets—ideally providing cell-type-specific replication timing and validated ERFS coordinates—become available, our model stands ready as a practical tool to assess how misfits at ERFS compare to CFSs and other fragile regions.

Although the model does not incorporate detailed molecular mechanisms, regions with high origin firing rates were nonetheless strongly associated with active transcription, open chromatin, and promoter activity (Fig. 5). These findings align with established knowledge, validating the model and underscoring its robustness. Notably, many misfit regions overlap with known fragile sites or distinct genomic locations, leading to the hypothesis that the model can refine the definition of fragile sites, distinguishing smaller, more nuanced regions of fragility, or even identifying sites prone to replication stress. Such predictions highlight the model's utility in uncovering unexplored genomic vulnerabilities, warranting further experimental validation.

Our approach has various limitations. For instance, we assume that each origin fires independently of others, which may not capture the full complexity of origin licensing and activation (see Methods). However, this simplification allows the model to fit human Repli-seq data rapidly, making it a practical tool for genome-wide analyses. Even so, in reality, a multiplicity of factors (e.g., ORC, Cdc6, and MCM proteins) regulate complex pathways of origin licensing, while later checkpoints and stress response pathways influence cell-cycle progression[56]. Another limitation is that we take no account of higher-order genome structure, but could incorporate data from, for example, Hi-C[6] and the position of R-loops, hairpins and G-quadruplexes that are known to obstruct replication[51]. Furthermore, our model could highlight the relationship between origins and DNA break clusters, such as those found at timing transition regions, which are prone to replication-transcription conflicts and genome instability[57]. Additionally, because Repli-seq data represent population averages, our analysis does not capture potential heterogeneity at the single-cell level[58,59]. Future studies employing single-cell data could thus provide finer resolution of replication dynamics.

Another nuance is that, while our model is quite universal in its assumptions, applying it to organisms like *S. cerevisiae* (budding yeast), which have smaller genomes and precisely located origins[60], may require adjusted parameterisation. In particular, the radius of influence $R$ becomes more critical in a smaller genome where it can have a proportionally greater effect. Outside of autonomously replicating sequences (ARS), the model is expected to assign very low firing rates by default. In Supplementary Note 2.6, we demonstrate the application of our framework to yeast, where the model successfully recovers >86% of known origin locations using only timing data, supporting our hypothesis and highlighting the model's general applicability across eukaryotes. Further investigation in yeast will be presented in a future study.

An exciting application of the model involves exploring the impact of chemotherapies on replication dynamics, particularly those therapies that target the Replication Stress Response (RSR) pathway and its key signalling proteins. By simulating the inhibition of these proteins, the model could provide valuable insights into how these disruptions affect replication timing, origin firing, and potential cell death[61]. This could facilitate prediction of which combination chemotherapies might provide cost-effective approaches to optimise cancer treatments.

## Methods
### Modelling assumptions
Our model is built on several key assumptions. First, the firing time of an origin is modelled as an exponentially distributed random variable, independent of fork movement and of the firing times of other origins. Second, replication forks progress at a constant speed, regardless of the dynamics of origin firing. This constant speed assumption serves as a critical constraint when benchmarking wild-type replication. If fork speed were allowed to vary freely in space and time, it would be possible to adjust fork progression locally to match steep timing profiles, resulting in multiple equally valid solutions and reducing the model's predictive power.

The origin firing rate encompasses origin licensing and activation, plus contributions of all other proteins and pathways within this process. While a strong assumption, it is justified by the fact that firing rates effectively capture the collective outcome of all these underlying processes without explicitly representing molecular detail. This makes the model both tractable and capable of producing accurate genome-wide predictions. We further sub-divide the genome into 1 kb intervals (sites), and assign to each a non-zero firing rate determined by a governing equation that links timing with firing. This resolution offers a balance between computational efficiency and biological realism. Although any site is a potential origin, our fitting algorithm can effectively turn off potential origins by assigning them a suitably low firing rate.

We also intentionally omit finer details of strand synthesis. In particular, we do not distinguish between leading and lagging strands, nor do we model the formation and joining of Okazaki fragments. By concentrating on the fundamental kinetics driving replication, we gain a clearer understanding of how origin firing rates shape replication timing without introducing unnecessary complexity.

We now present the main framework leading to Eq. (1).

## Mathematical modelling of replication

Consider a DNA molecule with $n$ discrete genomic loci, where each locus can potentially act as an origin that fires at rate $f$ to initiate a fork that progresses bidirectionally with speed $v$, typically measured in kilobases per minute (kb/min). We aim to determine the average time required for a site to either initiate replication or to be passively replicated by an approaching fork (i.e., its expected replication time). Initially, we assume that all origins fire at the same rate, $f$, but later relax this assumption to allow for variations in firing rates across different origins. In addition, by considering a sufficiently large chromosome, we ensure that effects of chromosomal ends are negligible. Nonetheless, the framework can easily be extended to account for such effects, though they are not critical for the broader analysis.

**Expected time of replication.** Let $T$ be the time a site takes to fire or be passively replicated by a fork. We assume initially that all origins fire at the same rate, $f$. One may think of $T$ as an explicit function of origin firing times $A_i$, where $A_i \overset{\text{iid}}{\sim} \text{Exp}(f)$. In particular, $\mathbb{E}[A_i] = 1/f$. We index each site by its distance from the origin of interest, given by $|i|$. Notice that $i = 0$ corresponds to the focal origin, and $v$ is interpreted as the number of replicated sites per time unit. We have

$$T = \min_i \{A_i + |i|/v\} \tag{2}$$

since it takes time $|i|/v$ for a fork initiated at site $i$ to reach the origin of interest. Next, we compute the cumulative distribution function. The minimum in Eq. (2) is greater than some $t$ if all terms are, which occurs with probability

$$P(T > t) = \prod_i \min\{1, \exp(-f(t - |i|/v))\} \tag{3}$$

since $A_i > 0$ and $A_i \sim^{\text{iid}} \text{Exp}(f)$. Hence, the expectation of replication time for any one site is given by

$$\mathbb{E}[T; n] = \int_0^\infty \prod_i \min\{1, \exp(-f(t - |i|/v))\} \, dt \tag{4}$$

where the product is taken over all $n$ sites. This integral can be partitioned across each interval for which $|i| \le vt \le |i + 1|$. Within these intervals, integrands adopt the form $ae^{-bt}$, thereby permitting analytical evaluation. In the general case, the result depends on the parity of $n$. See Supplementary Note 1.1 for an explicit expression of $\mathbb{E}[T; n]$.

As $n \to \infty$, a general expression of the expected replication time for each origin can be written as

$$\mathbb{E}[T; \infty] \equiv \frac{1}{f} \sum_{k=0}^\infty \frac{e^{-fk^2/v} - e^{-f(k+1)^2/v}}{2k+1}. \tag{5}$$

With $v = 1.4$ kb/min[31], Fig. 6a shows the dynamics of $\mathbb{E}[T; n]$ for increasing values of $n$. By relating Eq. (5) to the family of theta and Dawson functions, the following approximation holds (see Supplementary Note 1.2 for a detailed proof)

$$\mathbb{E}[T; \infty] \simeq \frac{1}{2} \sqrt{\frac{\pi}{fv}}. \tag{6}$$

Provided replication timing data $\{T_j\}_{1 \le j \le n}$, we have the following inversion

$$f_j \simeq \frac{\pi}{4v} T_j^{-2} \tag{7}$$

which provides a first estimate for the intrinsic firing rate of an origin, given its time of replication. Note that Eq. (7) is an approximation under the specific assumption that firing rates are uniformly constant across the genome, a simplification that, intriguingly, offers a reasonably accurate initial estimate for the firing rate distribution in most instances. The fidelity of this approximation is closely tied to fork speed $v$ and the average of the timing dataset, topics that will be elaborated subsequently.

**A generalisation.** Experimental data support the idea that different origins fire at different rates[62]. While our introductory argument assumes a constant firing rate $f$ across the genome, we should, in general, expect $A_i \sim \text{Exp}(f_i)$. Then, the replication time definition in Eq. (2) should include the site-specific indexation, for $1 \le j \le n$, as follows

$$T_j = \min_i \{A_i + |i - j|/v\} \tag{8}$$

with indexes congruent modulo $n$, that is, $|i - j| \in \mathbb{Z}/n\mathbb{Z}$ (see Supplementary Note 1). Following a similar argument, the general expression for $\mathbb{E}[T_j; \infty]$, with general firing rates $\{f_i\}$, is given by

$$\mathbb{E}[T_j; \infty] = \sum_{k=0}^\infty \frac{e^{-\sum_{|i| \le k}(k-|i|)f_{j+i}/v} - e^{-\sum_{|i| \le k}(k+1-|i|)f_{j+i}/v}}{\sum_{|i| \le k} f_{j+i}}. \tag{9}$$

When $f_j = f$, $\forall j$, Eq. (9) is reduced to Eq. (5). While Eq. (9) holds true for an infinitely large genome, in practical terms this series can be limited to $0 \le k \le R < n/2$, for some large enough $R$, leading to Eq. (1). This parameter represents the radius of replication influence: the distance within which neighbouring origins $\{j - R, \ldots, j - 1, j + 1, \ldots, j + R\}$ are assumed to affect the timing of a focal origin $j$. In other words, while every firing origin does theoretically affect replication timing at any other location, this effect decays rapidly with distance from the origin of interest $j$. Numerically, the finite version of Eq. (9) should mimic the average replication timing obtained from computational simulations, and it will be crucial in solving the fitting problem efficiently. Ideally, we would like to compute the rates $\{f_j\}_{1 \le j \le n}$ as a function of the expectation of $T_j$. Our goal is then to find a solution to Eq. (9), given data on $\{\mathbb{E}[T_j; n]\}$, for large $n$. Alternative frameworks inspired by the analogy between DNA replication and crystal growth have been previously explored by Jun, Bechhoefer, and Rhind[22,23,63], revealing other relevant replication metrics, such as inter-origin distances[64]. Our formulation extends these approaches by estimating origin firing rates from discrete replication timing data across the entire human genome, which is discussed next.

## Replication timing data

Replication timing data were sourced and processed from two key databases: the Encyclopedia of DNA Elements (ENCODE[65,66]) and high-resolution Repli-seq from Zhao et al.[28]. To ensure data consistency and reliability, extensive filtering and scaling steps were performed on all data sets. We analyse data from: HUVEC (human umbilical vein endothelial cells), HeLa-S3 (clonal derivative of the parent HeLa, an immortalised cervical cancer line), BJ (normal skin fibroblast), IMR90 (lung fibroblast), K562 (lymphoblast cells), GM12878 (lymphoblastoid line), HepG2 (hepatocellular carcinoma line), MCF-7 (breast cancer line), HCT116 (colorectal carcinoma line), plus H1 and H9 (embryonic stem cell lines). Data for HUVEC, HeLa, BJ, IMR90, K562, GM12878, HepG2, and MCF-7 cells were obtained from the ENCODE database using the GRCh37 (hg19) human genome assembly[65,66], while data for HCT116, H1, and H9 cells were sourced from high-resolution Repli-seq, using the GRCh38 (hg38) assembly[28].

Regarding ENCODE Repli-seq, timing data from each cell line were analysed across 6 cell cycle fractions: G1/G1b, S1, S2, S3, S4, and G2, given as a wavelet-smoothed signal to generate a continuous portrayal

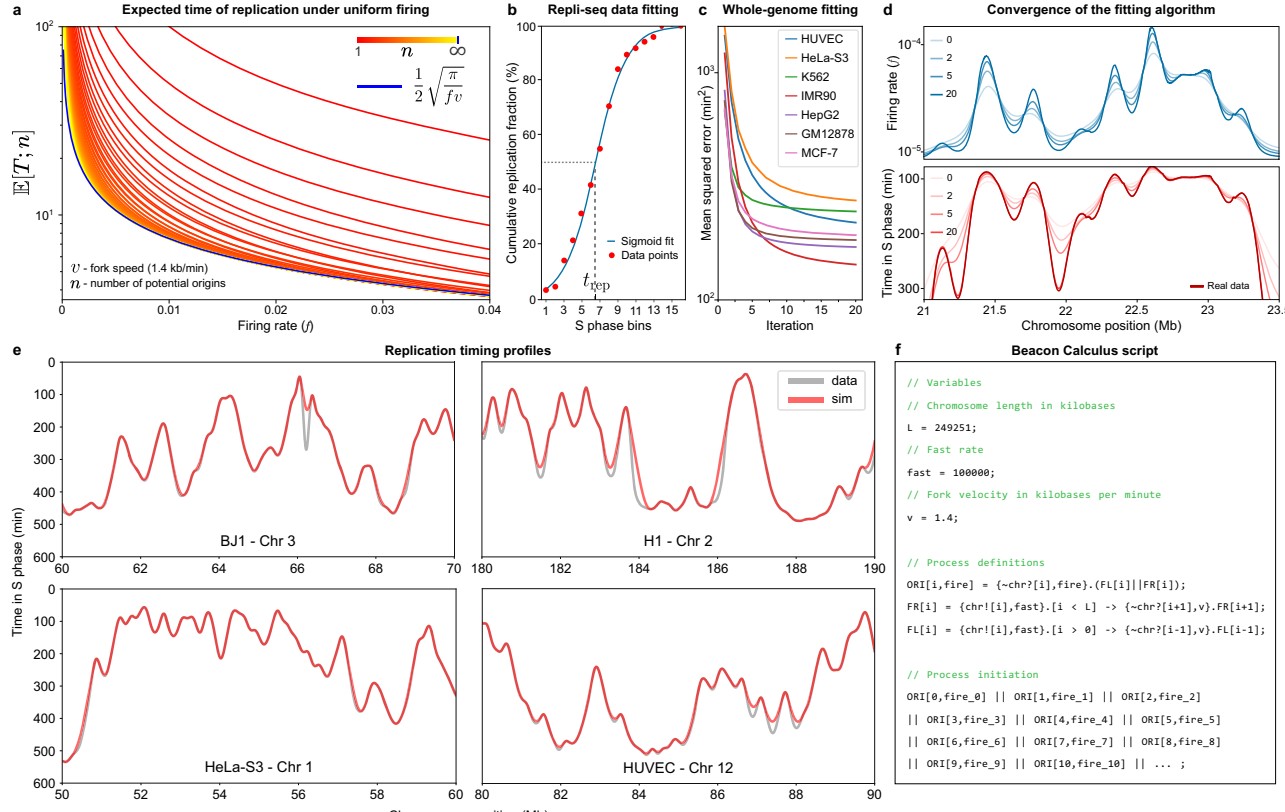

**Fig. 6 | Fitting the model. a** Replication asymptotics under uniform firing: logarithmic plot of the expected replication time, $\mathbb{E}[T; n]$, as a function of the firing rate, $f$, and the number of potential origins, $n$ (spaced at 1 kb intervals), for $1 \leq n < \infty$, with $v = 1.4$ kb/min. As $n \to \infty$, $\mathbb{E}[T; n]$ approximates an inverse power law (blue). **b** Curve fitting for cumulative replication in S phase. Red markers depict example data points from a high resolution Repli-seq heatmap that shows the cumulative percentage of completed replication across 16 S phase bins. The blue line is the curve fitted to this data, while the dashed grey line indicates the median replication time, $t_{rep}$ (the instant in S phase when 50% of replication is achieved across the cell population). **c** Whole-genome mean squared error between simulated timing profiles and real data for 7 cell lines, in min². Fitting each line took ~3 min on a HPC

platform (one CPU). **d** Progression of the fitting algorithm over 20 iterations for chromosome 2 in the BJ line on firing rates (above), with iteration 0 corresponding to the initial inverse power law estimate, given by Eq. (7), and the corresponding timing profile (below). **e** Observed (Repli-seq) timing against the simulated profiles for different lines and genomic regions. **f** Model written in the Beacon Calculus process algebra. Origin firing processes take their location, `i` (1-kb resolution), and firing rate `fire`, as parameters, triggering two replication fork processes, `FL` (left-moving) and `FR` (right-moving). Replication terminates when all locations have been replicated. The simulation begins by invoking the `ORI` processes, where `fire_i` corresponds to the firing rate values for each origin `i`, as determined by fitting Eq. (1).

---

of replication across the genome[67]. Importantly, we rescaled the original wavelet signal, initially normalised from 0 to 100, by a factor of 6 to better align with an approximately 8-hour S phase. Following standard Repli-seq methods, we applied a sigmoidal fit to the cumulative replication fraction, $F_{rep}$, to determine replication timing according to Zhao et al.[28]. We consider the median replication time, $t_{rep}$, defined as the bin value $t$ where $F_{rep}(t) = 50\%$, indicating that half of the cell population has completed replication (Fig. 6b). Although Eq. (9) theoretically represents the mean replication timing, it aligns closely with the median observed in Repli-seq data, as replication timing distributions generally exhibit a near-symmetric sigmoidal pattern. Additionally, the median is more robust to experimental noise and outliers, making it a practical and reliable measure in high-throughput experiments. Although recent studies have determined telomere timing data[68], we do not incorporate them into our analysis. Repli-seq data shows consistent patterns across different cell lines. We present representative results from multiple lines, but specific analyses may be more suitable for certain cases, depending on the availability and quality of the data. Although regions with repetitive sequences or low complexity are often poorly mapped using Repli-seq data[28,65], these regions account for ~20% of the genome and show only a weak correlation with high-misfit regions (phi coefficient = 0.21). Therefore, we

retain this data in our analysis, as its impact is minimal (Supplementary Note 2.3).

## Fitting algorithm

Equation (9) establishes a continuous, monotonic relationship between each firing rate, $f_j$, and its corresponding replication time, $\tilde{T}_j$. Our aim is to infer the set of firing rates $\{f_j\}$ from experimentally measured timing data $\{T_j\}$. Rather than relying on large-scale simulations, we employ an iterative procedure that leverages the monotonic relationship in Eq. (9): each firing rate is updated so as to minimize the difference between $\tilde{T}_j$ (predicted via Eq. (9)) and $T_j$ (obtained from Repli-seq data). In practice, for large $n$, Eq. (9) provides a good approximation of $\tilde{T}_j$. We initialize each site $j$ by

$$f_j(0) = \frac{\pi}{4v} T_j^{-2}, \tag{10}$$

then iteratively refine its firing rate according to

$$f_j(k+1) = f_j(k) \left( \frac{\tilde{T}_j(k)}{T_j} \right)^{\alpha}, \tag{11}$$

where $\tilde{T}_j(k)$ is the predicted replication time at iteration $k$, and $T_j$ is the experimentally observed timing. The exponent $\alpha$ governs how strongly firing rates respond to each site's misfit, behaving much like a fixed-point iteration or an inexact gradient descent. Numerical experiments suggest $\alpha = 2$ provides a robust balance between speed of convergence and numerical stability, but other choices of $\alpha$ are feasible if the data require finer control or if a gradient-based approach is desired. Every 1 kb segment is treated initially as a potential origin, but the algorithm's updates naturally drive most firing rates to negligible values, reflecting the selective activation of origins in the genome. The radius of neighbouring influence, $R$, may be refined for optimisation. We track convergence by measuring each site's fit error, defined as the squared difference $(T_j - \tilde{T}_j(k))^2$, in min². Because the method directly leverages the convolution-like form of Eq. (9), it avoids repeating large-scale simulations. This efficiency means that fitting 3.2 million sites per human genome can typically be done within a few minutes on a single Intel Ice Lake CPU (Fig. 6c–e). Thus, although we rely on simple iterative corrections, the monotonic structure of $\tilde{T}_j$ with respect to $f_j$ ensures the scheme converges reliably, provided $\alpha$ and other parameters remain moderate. We have added a more detailed discussion of this algorithm, including its convolution interpretation, in Supplementary Note 2.2.1.

## Simulations

To simulate replication, we use Beacon Calculus (`bcs`)[27], a process algebra designed for simulating biological systems. In this framework, each component of the system is treated as a process capable of executing certain actions, each governed by an exponential rate. The simulation uses a modified Gillespie algorithm, allowing multiple processes to run in parallel, a property that is especially important for modelling DNA replication, where many events occur concurrently. In our case, we represent replication with three processes: replication origin firing (`ORI`), and passive replication by left- (`FL`), and right-moving forks (`FR`). Each process is associated with a specific position at a given resolution on a chromosome of length `L`, and origins have an additional parameter, the firing rate, `fire`, or $f$ in our model (Fig. 6f).

In each `bcs` simulation, when a process is activated and its associated action is executed (i.e., when a random variable is realised), the time and location of that event is recorded. Our model operates at a 1 kb resolution, so each action is assigned to a specific 1 kb segment; consequently, the firing of an origin (`ORI`) or the passive replication by a fork (`FL` or `FR`) is registered as an event within that segment. All sites are treated as potential origins; however, the fitting algorithm effectively turns them on or off by assigning them high or near-zero firing rates, respectively. The model is flexible and can be adapted to different resolutions. In our current implementation, an origin is defined as corresponding to a 1 kb segment. This computational implementation is independent from the analytical approach described above, providing a means to confirm the closed-form mathematical analysis and explore additional replication features. Further details on the `bcs` formalism and its usage are discussed in Supplementary Note 2.1.

In `bcs`, $v$ is treated as the constant replication rate of a moving fork, i.e., the parameter of an exponential distribution governing the time required to passively replicate one site. This differs from the constant fork speed assumption underlying Eq. (9). Specifically, in the `bcs` case, the time $F_k$ required for a fork to replicate $k$ consecutive sites follows an Erlang($k$, $v$) distribution, meaning that $\mathbb{E}[F_{|i-j|}] = |i-j|/v$, which mirrors the approximation used in Eq. (8). Therefore, when averaged over a sufficiently large number of simulations, stochastic deviations in numerical simulations become negligible and they do not compromise the broader analysis or conclusions. To track the progress of replication, the model marks regions of the chromosome that have been replicated, allowing us to monitor replication dynamics accurately. In all `bcs` simulations, fork speed was set to 1.4 kb/min[31], and results were averaged over 500 simulations, with the radius of influence set to $R = 2000$ kb, as previously defined.

### Reporting summary

Further information on research design is available in the Nature Portfolio Reporting Summary linked to this article.

## Data availability

The replication timing data used in this study were obtained from the Encyclopedia of DNA Elements (ENCODE) using the GRCh37 (hg19) assembly[65,66] and from high-resolution Repli-seq aligned to the GRCh38 (hg38) assembly[28]. RNA-seq[18], ChIP-seq[19], and GRO-seq[20] datasets were also obtained from ENCODE. Additionally, GRO-seq data used in this study were accessed from the Gene Expression Omnibus (http://www.ncbi.nlm.nih.gov/geo/) under accession numbers GSE62046, GSE94872, and GSE60454. Fragile site locations were sourced from the publicly available HumCFS database[21], accessible at https://webs.iiitd.edu.in/raghava/humcfs/. All these data are publicly available.

## Code availability

The source code implementing the main fitting algorithm, together with the replication timing fit error and origin firing rate bedgraph files, is available at: https://github.com/fberkemeier/DNA_replication_model.git (version 1.0.0, https://doi.org/10.5281/zenodo.15337522)[69]. Beacon Calculus simulations were carried out using version 1.1.0 of `bcs`, accessible at https://github.com/MBoemo/bcs[27]. Supplementary Note 2 provides additional examples of `bcs` scripts and optimisation algorithms.

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

## Acknowledgements

We thank Prof. Sarah McClelland (Barts Cancer Institute, Queen Mary University of London) and Dr. Mathew Jones (Frazer Institute, University of Queensland) for their constructive and insightful feedback, which significantly improved the manuscript. We also thank all members of the Boemo lab for their helpful discussions and comments. This work was performed using resources provided by the Cambridge Service for Data Driven Discovery (CSD3), operated by the University of Cambridge Research Computing Service (https://www.csd3.cam.ac.uk), and supported by Dell EMC and Intel through Tier-2 funding from the Engineering and Physical Sciences Research Council (capital grant EP/T022159/1) and DiRAC funding from the Science and Technology Facilities Research Council (https://dirac.ac.uk). This work was made possible by the Leverhulme Trust Research Project Grant RPG-2022-028, which was successfully applied for and secured by M.A.B. Additional funding and career support were provided by a Rokos Postdoctoral Associate position at Queens' College Cambridge to F.B. and a fellowship at St John's College, Cambridge to M.A.B.

## Author contributions

The project was conceived by F.B. and M.A.B. F.B. was responsible for conceptualisation, methodology and formal analysis. F.B., M.A.B., and P.R.C. contributed to data interpretation. F.B. was responsible for writing the original draft. M.A.B. and P.R.C. reviewed and edited the manuscript. M.A.B. applied for and was awarded the funding.

## Competing interests

The authors declare no competing interests.
