## [Transparent Peer Review file · Nature Communications]

DNA replication timing reveals genome-wide features of transcription and fragility

Corresponding Author: Dr Francisco Berkemeier

Version 0:

Reviewer comments:

Reviewer #1

(Remarks to the Author)

This reviewer specializes in molecular biology and not in mathematical modeling. Therefore, my comments will primarily focus on the molecular biology perspective.

In this study, the authors developed a novel mathematical model for genome-wide simulations of DNA replication firing rates and replication timing. The model appears to effectively capture several replication dynamics observed in experimental data, including replication timing. Moreover, the authors suggest that regions where the simulated replication timing profiles deviate from experimental data, referred to as misfit regions, could predict vulnerable genomic loci, such as common fragile sites.

I believe the model has potential and could be a valuable tool for future research in the field of DNA replication. However, to strengthen the conclusions, additional analyses are necessary (see below). Furthermore, I have concerns about the extent to which this study provides new biological insights. In its current form, the manuscript may be more suitable for publication in a specialized journal.

Major Points:

1. Many conclusions seem to rely on qualitative analyses. For instance, the authors state that “Regions such as centromeres and telomeres, as well as most fragile sites often map to regions of high error, particularly during late S phase (Figures 5a-d).” However, it is unclear whether the “Time-error distribution” for these regions differs significantly from that of the whole genome or the corresponding chromosomes. Can the authors perform additional quantitative analyses, including statistical evaluations, to support these claims? Similarly, in Figure 5e, I believe the authors could quantitatively compare the observed data to a control dataset encompassing all genes, which would provide a more robust analysis.

2. In the discussion, the authors suggest that investigating early replicating fragile sites (ERFS), which are associated with highly transcribed genes, is an important topic for future research. I recommend conducting these analyses in the present study. If novel biological insights into ERFS can be obtained, they would significantly enhance the quality of the manuscript. For example, determining whether the misfit distribution pattern in ERFS resembles that of common fragile sites would be intriguing.

Minor point:

1. Figure 5h: What is the difference between light pink and white?

(Remarks on code availability)

Reviewer #2

(Remarks to the Author)

This paper proposes the promising idea of inferring the firing rates of replication origins from timing data by using a mathematical model

of the replication dynamics. One particularly interesting feature of this work is that it assumes no knowledge of the positions of the replication origins, and instead allows every position in the DNA to potentially have the ability to originate replication forks; in this, it is different from most models in this area. It also applies the results of the fitting to many different human cell-lines, and uncovers suggestive links between model mismatches and various types of replication stresses. So this work offers an interesting and promising new approach to DNA replication, and intriguing biological insights. For those reasons, I think it merits publication in Nature Communications, provided the issues below are satisfactorily addressed.

Main remarks/inquiries:

* In the conclusion, you say that the "assumptions" of your method would need to be "adjusted" in order for it to work with organisms with localised origins such as *S. cerevisiae*. Why is that? It seems to me that an organism like *S. cerevisiae* should be just one particular case of your algorithm, where most of the 1 kb segments would have zero (or very low) firing rate. That would in fact be a good independent test of your method - you should be able to infer the positions of at least the most active origins, which could be verified by the use of origin location databases that are available for that organism (and a few others). Why is it not possible to do that using your method? Whether the reason be some fundamental assumption the method makes that is broken, or the numerical iteration method does not perform well in that case, this should be clearly explained in the paper.

* The fitting algorithm in page 5 should be better motivated and explained: why is the iterative procedure described there expected to converge to the actual solution of the system of equations in eq. 8 (or a good approximation thereof)? That is not clear to me.

* Bottom of page 5: "... underscoring the algorithm's reliance on higher firing rates to achieve the best fit with theoretical expectations." What do you mean when you say that the fitting algorithm relies on higher firing rates? Does that imply that the algorithm has an intrinsic tendency to overestimate the firing rates? If so, why?

This is an important issue to understand, because this could challenge the relevance of the finding that "late-replicating regions consistently exhibited higher misfit rates". Could this be just a quirk of the fitting algorithm, and nothing to do with those regions being more prone to replication challenges? Is there some way to verify that that fitting algorithm does not introduce a bias favouring higher firing rates? For example, by running a simulation on a "virtual" chromosome where you prescribe firing rates, and seeing if your method rediscovers those? It would have to be a different kind of simulation than the one used in your fitting method. I think it is important to seriously address this, since so many of the paper's conclusions rely on the assumption that there is no such bias.

* In Fig 3g, you show distributions of inter-origin distances. But since each 1 kb DNA segment has a firing rate, how do you define what an "origin" is? Do you use a threshold? This should be explained.

* In page 2: "and fork movement as an exponentially distributed random variable (independent of origin firing and movement of other forks). We also assume a constant rate of fork movement throughout (no fork stalling at obstacles); then, forks advance smoothly until encountering another fork or chromosome end. This assumption avoids overfitting..."

I think this passage is misleading, because it suggests that the

forks move at constant speed ("constant rate of fork movement"), when in fact they have a constant hopping rate, which is something quite different. The term "the forks advance smoothly" is also vague and potentially misleading.

The last sentence in that passage is puzzling to me: "This assumption avoids overfitting". Why? Please explain this.

(Remarks on code availability)

Reviewer #3

(Remarks to the Author)

This paper develops a mathematical model to fit replication timing data and simulate key replication parameters, from which predicted replication profiles are generated and compared to the original timing data. A main focus is on misfits- regions in which the model deviates from the data. The authors analyze these regions and make some interesting biological inferences about the replication process.

While otherwise very well presented, the paper doesn't sufficiently and clearly explain the actual math that goes into their modeling. It remains unclear how the model is generated and how the predicted replication profiles are obtained. The authors should keep in mind that Nature Communications is mostly geared towards biologists rather than mathematicians. For instance, the results only mention "replication is simulated using Beacon Calculus (bcs), a concise process algebra ideal for concurrent systems." This is not clear and doesn't provide enough information for the typical reader to evaluate and understand the approach.

In particular, the authors need to justify that their model isn't circular: it start with replication timing data as input, and using those data predicts replication timing for the same samples, which are then compared to the original data. It would seem that any deviation between the two would solely represent the limitations of the model. In particular, late replicating regions are known to be less consistent, more noisy, and more influenced by repetitive sequences, GC content and other technical influences, so it wouldn't be a surprise to find misfits concentrated in late-replicating areas. They also find misfits concentrated near telomeres and centromeres, which again are regions known for relatively poor quality replication timing data due to repetitive sequence and smoothing artifacts. Late replication is correlated with chromosome fragility, overall low gene density but nonetheless an enrichment of long genes, and other properties some of which the authors report as results. So it remains possible that a large fraction of the reported findings are due to model limitations related to data quality in late-replicating regions and to correlations of biological attributes with late replication.

Most of the biological insights reported in the paper are not novel, and it is not convincing that the model predicts them as biological rather than technical outcomes. Differently stated, it is not convincing that the model quantifies replication stress and other attributes of replication dynamics.

The authors disclose that "Misfits also vary between cell lines, with HCT displaying a distinct pattern likely due to differences in data processing". This would be consistent with technical artifacts influencing at least some of the results.

In the discussion, the authors cite Gindin et al, 2014 with regards to incorporating Hi-C data, however Gindin et al did not use Hi-C data in their paper.

(Remarks on code availability)

Version 1:

Reviewer comments:

Reviewer #1

(Remarks to the Author)

It is unfortunate that it is difficult to readily analyze the ERFs, but the authors have made a reasonable attempt to address the points raised by this reviewer, improving the quality of the paper. Therefore, I would like to support its publication in Nature Communications.

(Remarks on code availability)

Reviewer #2

(Remarks to the Author)

The authors answered most comments to my satisfaction. However, they did not test their method on *S. cerevisiae* as suggested in my previous report. I believe this is an important "control experiment" of their method: it should be able to recover the location of the most active origins in that organism, which are well known. This would also help with concerns about the circularity of the method, as expressed by one other referee. If that is done, I would recommend the paper for publication.

(Remarks on code availability)

Reviewer #3

(Remarks to the Author)

I appreciate the authors' efforts in responding to my (and the other reviewers) comments. While the manuscript has been improved, I still find it more appropriate for a more specialized journal; my main concern, that many of the results are due to noise in late-replicating regions and close to chromosome ends/gaps, remains. I have not seen a reproduction of the results when controlling for the late replication timing of fragile sites and long genes for instance, and the biological novelty of the findings remains limited. While the analysis and methods are sophisticated and the paper very well presented, it falls short of providing a sufficient addition to the understanding of DNA replication biology.

(Remarks on code availability)

Version 2:

Reviewer comments:

Reviewer #2

(Remarks to the Author)

The authors' responses address my only remaining concern, so I recommend that the paper be accepted for publication.

(Remarks on code availability)

Reviewer #3

(Remarks to the Author)

The authors have addressed all of my concerns, thank you.

(Remarks on code availability)

Reviewer #1

Comment 1.1

Many conclusions seem to rely on qualitative analyses. For instance, the authors state that “Regions such as centromeres and telomeres, as well as most fragile sites often map to regions of high error, particularly during late S phase (Figures 5a-d).” However, it is unclear whether the “Time-error distribution” for these regions differs significantly from that of the whole genome or the corresponding chromosomes. Can the authors perform additional quantitative analyses, including statistical evaluations, to support these claims?

We thank the reviewer for emphasizing the need for additional quantitative assessments. Our discussion of fragile sites is intended to illustrate how our model can highlight regions of potential replication stress, such as large genes within fragile sites, rather than to claim that all fragile sites display uniform time-error profiles. We have clarified this in the main text. To support this, we performed targeted quantitative analyses in HCT116 cells, where confirmatory data on fragile site expression are available (Zhao et al., 2020; Boteva et al., 2020). Specifically, we compared replication misfit distributions between fragile and non-fragile loci, both across the full dataset and within a high-error subset, thereby identifying several genes with pronounced misfits. Detailed statistical analyses are provided in the newly added Section 2.4 in Supplementary Information. Overall, our findings demonstrate that while a global correlation between misfits and fragile sites is not expected, given the localized, cell-type-specific nature of these features, fragile sites often exhibit statistically significant differences in replication misfits, particularly on specific chromosomes. This supports the utility of our model as a robust framework for identifying candidate regions for further experimental investigation. We introduced the following revisions to the manuscript:

Page 5 (in *Results: Fragile sites and long genes*):

“Fragile sites are also known to be cell-type specific (Letessier et al., 2011; Le Tallec et al., 2011), yet our analysis of 11 lines, with H1 cells serving as one illustration, suggests that core fragility-misfit correlations are relatively consistent even if the degree of disruption varies. A more detailed case study of HCT116, where confirmatory data on fragile site expression is available (Boteva et al., 2020), is included in the Supplementary Information (SN2.4), highlighting how individual chromosomes and cell-line dependent features can shape replication stress at fragile sites. Taken together, these findings reinforce that fragile sites often correlate strongly with replication misfits, though not uniformly across the genome or in every cell type. By pinpointing likely hotspots of replication stress, our model provides a powerful framework for guiding experimental follow-up.”

Page 7 (in *Discussion*):

“We also performed a statistical assessment of misfit distributions in HCT116, a cell line with robust confirmatory data on fragile site expression (Boteva et al., 2020). Our results indicate that although fragile sites in HCT116 frequently show statistically significant differences in replication misfits, especially on certain chromosomes, this pattern is not uniform across the entire genome. Further targeted studies could help clarify how cell-line-specific factors shape localized vulnerability within a broader replication error landscape. At the same time, by pinpointing potential replication stress hotspots, our model provides a valuable foundation for deeper experimental investigations into the molecular underpinnings of fragility. Researchers with access to cell type-matched Repli-seq and fragility data could refine this framework to achieve more specific insights.”

Pages 6-8 (in *Supplementary Information*):

We added an entire new section, '2.4. Fragility analysis in HCT116', discussing the detailed statistical analyses of misfit distributions in fragile sites on HCT116 cells.

Comment 1.2

Similarly, in Figure 5e, I believe the authors could quantitatively compare the observed data to a control dataset encompassing all genes, which would provide a more robust analysis.

We agree with the reviewer's suggestion, as this additional comparison more clearly underscores the relative misfit fraction in genic regions compared to genes at fragile sites, thereby offering a stronger connection to the results discussed in Figure 5 (6 in the previous version of the manuscript, as the Methods section has now been moved to follow the Discussion in accordance with Nature Communications' formatting requirements). We have now included the misfit fraction for all genes in Figure 4f (previously 5f, which focuses on gene-level data). Additionally, we have added the misfit fraction across all fragile sites (from the HumCFS database) in Figure 4e (previously 5e). We've adapted the figure caption in light of these changes:

Page 6 (in *Results: Fragile sites and long genes*; caption of Figure 4):

"(e) Misfit distribution for common (blue) and rare (pink) fragile sites, compared with the total fragile site misfit fraction (grey). Top: length (in Mb) of continuous misfit regions. Bottom: normalised misfit fraction at different sites. (f) Misfit fraction analysis of whole-genome genic regions, and at the largest genes within fragile sites (normalised)."

Comment 1.3

In the discussion, the authors suggest that investigating early replicating fragile sites (ERFS), which are associated with highly transcribed genes, is an important topic for future research. I recommend conducting these analyses in the present study. If novel biological insights into ERFS can be obtained, they would significantly enhance the quality of the manuscript. For example, determining whether the misfit distribution pattern in ERFS resembles that of common fragile sites would be intriguing.

We appreciate the reviewer's suggestion to explore early replicating fragile sites (ERFS) more extensively. Indeed, ERFS represent an intriguing area for deeper investigation. However, publicly available data on ERFS in mammalian cells are relatively sparse compared to the more robust datasets for common fragile sites, especially in human cell lines. Existing references on ERFS (e.g., Barlow et al. (2013) in mouse B cells, Pentzold et al. (2018) in avian cells) do not provide comprehensive, high-resolution ERFS coordinates in human systems, and their cross-species relevance may be constrained by differences in genome organization and replication programs. Furthermore, mapping ERFS from mouse (mm9 or mm10) onto our human timing datasets (hg18 or hg38) can produce confounding results, given established differences in replication programs across species and cell types. For now, we emphasize in our revised Discussion that our model stands ready to integrate ERFS data once suitable cell-type specific annotations are accessible, and briefly discuss a potential applicability with work from Tubbs et al. (2018). In order to acknowledge this, we have introduced the following revisions to the manuscript:

Page 7 (in *Discussion*):

“We also note that early-replicating fragile sites (ERFS), often linked to highly transcribed genes (Barlow et al., 2013), represent another compelling avenue for future work. For instance, the study by Tubbs et al. (2018) demonstrates that poly(dA:dT) tracts can precipitate replication fork collapse in both early- and late-replicating domains, suggesting that similar sequence features may underlie fragility across diverse replication windows. However, robust, high-resolution data on ERFS in human cell lines remain limited; most existing datasets come from mouse (Barlow et al., 2013) or avian cells (Pentzold et al., 2018), and cross-species mapping (e.g., via LiftOver) does not reliably capture species-specific replication landscapes. Once comprehensive human datasets—ideally providing cell-type-specific replication timing and validated ERFS coordinates—become available, our model stands ready as a practical tool to assess how misfits at ERFS compare to CFSs and other fragile regions.”

Comment 1.4

Figure 5h: What is the difference between light pink and white?

Light pink indicates genes overlapping rare fragile sites, while white and grey represent genes that fall outside any fragile site (either common or rare, according to the HumCFS database. For improved clarity, we have enhanced the legend in the top-right corner, removed the grey bands and added reference lines in Figure 4h (5h in previous manuscript version).

Reviewer #2

Comment 2.1

*In the conclusion, you say that the "assumptions" of your method would need to be "adjusted" in order for it to work with organisms with localised origins such as *S. cerevisiae*. Why is that? It seems to me that an organism like *S. cerevisiae* should be just one particular case of your algorithm, where most of the 1 kb segments would have zero (or very low) firing rate. That would in fact be a good independent test of your method - you should be able to infer the positions of at least the most active origins, which could be verified by the use of origin location databases that are available for that organism (and a few others). Why is it not possible to do that using your method? Whether the reason be some fundamental assumption the method makes that is broken, or the numerical iteration method does not perform well in that case, this should be clearly explained in the paper.*

We thank the reviewer for pointing this out and suggesting that our text could have been clearer. Indeed, our method can be applied to smaller genomes such as *S. cerevisiae*. The notion of “adjusting assumptions” mainly refers to practical parameters, particularly the radius of influence R discussed in the main text, which become more critical in a smaller genome. We appreciate this opportunity to clarify and have updated the main text accordingly to emphasize that *S. cerevisiae* is indeed within the scope of our approach. Rather than presenting it as a limitation, we change the phrasing to acknowledge the reviewer’s comment:

Page 8 (in *Discussion*):

*“Another nuance is that, while our model is quite universal in its assumptions, applying it to organisms like *Saccharomyces cerevisiae*, which have smaller genomes and precisely located origins, may require adjusted parameterisation. In particular, the radius of influence R becomes more critical in a smaller*

genome where it can have a proportionally greater effect. Outside autonomously replicating sequences, one would assign very low firing rates.”

Comment 2.2

The fitting algorithm in page 5 should be better motivated and explained: why is the iterative procedure described there be expected to converge to the actual solution of the system of equations in eq. 8 (or a good approximation thereof)? That is not clear to me.

The key point is that Eq. 9 (previously Eq. 8) defines an inverse relationship between each firing rate f_j and the corresponding replication time \tilde{T}_j . The iterative procedure leverages this by repeatedly correcting f_j in proportion to how much the predicted replication time diverges from the observed Repli-seq timing. Conceptually, it drives firing rates in the direction that reduces the overall misfit. We have verified empirically that the algorithm converges reliably if the updates remain moderate. The main text has been revised to clarify these points:

Pages 10-11 (in *Methods: Fitting algorithm*):

“Eq. (9) establishes a continuous, monotonic relationship between each firing rate, f_j , and its corresponding replication time, \tilde{T}_j . Our aim is to infer the set of firing rates f_j from experimentally measured timing data T_j . Rather than relying on large-scale simulations, we employ an iterative procedure that leverages the monotonic relationship in eq. (9): each firing rate is updated so as to minimize the difference between \tilde{T}_j (predicted via eq. (9)) and T_j (obtained from Repli-seq data). In practice, for large n , eq. (9) provides a good approximation of \tilde{T}_j . We initialize each site j by

$$f_j(0) = \frac{\pi}{4v} T_j^{-2}$$

then iteratively refine its firing rate according to

$$f_j(k+1) = f_j(k) \left(\frac{\tilde{T}_j(k)}{T_j} \right)^\alpha$$

where \tilde{T}_j is the predicted replication time at iteration k , and T_j is the experimentally observed timing. The exponent α governs how strongly firing rates respond to each site’s misfit, behaving much like a fixed-point iteration or an inexact gradient descent. Numerical experiments suggest $\alpha = 2$ provides a robust balance between speed of convergence and numerical stability, but other choices of α are feasible if the data require finer control or if a gradient-based approach is desired. Every 1 kb segment is treated initially as a potential origin, but the algorithm’s updates naturally drive most firing rates to negligible values, reflecting the selective activation of origins in the genome. The radius of neighbouring influence, R , may be refined for optimisation. We track convergence by measuring each site’s fit error, defined as the squared difference $(T_j - \tilde{T}_j)^2$, in min^2 . Because the method directly leverages the convolution-like form of eq. (9), it avoids repeating large-scale simulations. This efficiency means that fitting 3.2 million sites per human genome can typically be done within a few minutes on a single Intel Ice Lake CPU (Figures 6c-e). Thus, although we rely on simple iterative corrections, the monotonic structure of \tilde{T}_j with respect to f_j ensures the scheme converges reliably, provided α and other parameters remain moderate. We have added a more detailed discussion of this algorithm, including its convolution interpretation, in SN2.2.1.”

Comment 2.3

Bottom of page 5: "... underscoring the algorithm's reliance on higher firing rates to achieve the best fit with theoretical expectations." What do you mean when you say that the fitting algorithm relies on higher firing rates? Does that imply that the algorithm has an intrinsic tendency to overestimate the firing rates? If so, why? This is an important issue to understand, because this could challenge the relevance of the finding that "late-replicating regions consistently exhibited higher misfit rates". Could this be just a quirk of the fitting algorithm, and nothing to do with those regions being more prone to replication challenges? Is there some way to verify that that fitting algorithm does not introduce a bias favouring higher firing rates? For example, by running a simulation on a "virtual" chromosome where you prescribe firing rates, and seeing if your method rediscovers those? It would have to be a different kind of simulation than the one used in your fitting method. I think it is important to seriously address this, since so many of the paper's conclusions rely on the assumption that there is no such bias.

We thank the reviewer for raising this important point. The phrase "reliance on higher firing rates to achieve the best fit" may have been misleading and we are grateful for the opportunity to improve this. We did not mean to imply that the algorithm systematically inflates those rates. Rather, in regions where the timing profile is steep (e.g., where early-replicating regions transition to late-replicating regions), there is an upper bound on the magnitude of the timing curve gradient imposed by the assumption of a constant rate of fork movement. This is best illustrated by thinking about the timing curve of a single-origin system: there will be a sharp point at the origin and the gradient elsewhere will be determined by the rate of fork movement. Adding additional origins at any firing rate can only decrease the magnitude of the curve's gradient. Through the fitting procedure, our model highlights regions where replication happens later than would be expected under our model which can only happen when forks stall or slow beyond the rate specified in our model. We have then made the following adjustment, and a new section in Supplementary Information with the illustration of the one-origin system described above:

Page 3 (in *Results: Predicting genome-wide replication*):

"We focus on the regions with high misfit error (shaded in yellow and red in Figure 2b.i), where the assumption of constant fork speed in equation (1) leads the model to predict earlier replication times than are observed experimentally. Because a constant speed imposes an inherent upper bound on how quickly replication can transition from early to late (see SN2.2.2), any steeper or delayed transitions—potentially arising from fork stalling or local inefficiencies—remain unmatched by the model. These high-misfit zones thus highlight loci where forks appear to slow or stall beyond our simplified assumptions, flagging potential replication stress hotspots for more detailed study."

Pages 5 (in *Supplementary Information*):

We added an entire new section, '2.2.2. Effects of fork speed on replication timing misfits'.

Comment 2.4

In Fig 3g, you show distributions of inter-origin distances. But since each 1 kb DNA segment has a firing rate, how do you define what an "origin" is? Do you use a threshold? This should be explained.

We thank the reviewer for raising this important point. Since the precise locations of replication origins in the human genome are not fully understood, our initial condition is agnostic: every 1 kb segment is treated as a potential origin. Our fitting algorithm then effectively "activates" or "deactivates" these segments by

assigning them high or near-zero firing rates, respectively. In our simulation, each event—whether it is an origin firing (ORI) or passive replication by a fork (FL or FR)—is recorded for its 1 kb segment. Those segments where an event occurs are then defined as origins, and the inter-origin distance distribution is derived empirically from these events. Further details on this approach are provided in the Methods (Simulations) and Supplementary Note 2.1.

In order to clarify this, we have introduced the following revisions to the manuscript:

Page 11 (in *Methods: Simulations*):

“In each bcs simulation, when a process is activated and its associated action is executed (i.e. when a random variable is realised), the time and location of that event is recorded. Because our model operates at a 1 kb resolution, each action is assigned to a specific 1 kb segment; consequently, the firing of an origin (ORI) or the passive replication by a fork (FL or FR) is registered as an event within that segment. All sites are treated as potential origins; however, the fitting algorithm effectively turns them “on” or “off” by assigning them high or near-zero firing rates, respectively. The model is flexible and can be adapted to different resolutions. In our current implementation, an origin is defined as corresponding to a 1 kb segment. This computational implementation is independent from the analytical approach described above, providing a means to confirm the closed-form mathematical analysis and explore additional replication features. Further details on the bcs formalism and its usage are discussed in SN2.1.”

Comment 2.5

In page 2: “and fork movement as an exponentially distributed random variable (independent of origin firing and movement of other forks). We also assume a constant rate of fork movement throughout (no fork stalling at obstacles); then, forks advance smoothly until encountering another fork or chromosome end. This assumption avoids overfitting...”. I think this passage is misleading, because it suggests that the forks move at constant speed (“constant rate of fork movement”), when in fact they have a constant hopping rate, which is something quite different. The term “the forks advance smoothly” is also vague and potentially misleading. The last sentence in that passage is puzzling to me: “This assumption avoids overfitting”. Why? Please explain this.

We agree that the original phrasing could be misleading and appreciate the reviewer’s concern. We have clarified that our mathematical model assumes constant fork speed, and we have revised the text accordingly. We apologise for this lapse. The following revisions have been incorporated into the manuscript:

Page 8 (in *Methods: Modelling assumptions*):

“Our model is built on several key assumptions. First, the firing time of an origin is modelled as an exponentially distributed random variable, independent of fork movement and of the firing times of other origins. Second, replication forks progress at a constant speed, regardless of the dynamics of origin firing. This constant speed assumption serves as a critical constraint when benchmarking wild-type replication. If fork speed were allowed to vary freely in space and time, it would be possible to adjust fork progression locally to match steep timing profiles, resulting in multiple equally valid solutions and reducing the model’s predictive power.”

Reviewer #3

Comment 3.1

While otherwise very well presented, the paper doesn't sufficiently and clearly explain the actual math that goes into their modeling. It remains unclear how the model is generated and how the predicted replication profiles are obtained. The authors should keep in mind that Nature Communications is mostly geared towards biologists rather than mathematicians. For instance, the results only mention "replication is simulated using Beacon Calculus (bcs), a concise process algebra ideal for concurrent systems." This is not clear and doesn't provide enough information for the typical reader to evaluate and understand the approach.

We thank the reviewer for the constructive feedback. Recognizing that *Nature Communications* is primarily read by biologists, we have streamlined the main text to present only the essential steps and equations while relegating the full mathematical derivation to Supplementary Note 1 ("Mathematical Notes"), where we detail the proof of the expected replication time for the interested reader. In the main text, we first illustrate the simpler case of uniform firing before extending to the general model, to gradually guide the reader to the main techniques used here. In addition, we have expanded the "Simulations" section of the Methods to more clearly define our use of Beacon Calculus (bcs) for modelling replication. This additional detail explains what we mean by processes in bcs and why this framework is particularly well suited for capturing the concurrent nature of DNA replication. For readers interested in further technical details, Supplementary Note 2.1 ("Beacon Calculus Model") provides a deeper discussion with an example. Importantly, we emphasize that our novel methods are independent of the computational simulations, which serve as a powerful tool to confirm and explore additional replication features. The following revisions have been incorporated into the manuscript:

Page 11 (in *Methods: Simulations*):

"To simulate replication, we use Beacon Calculus (bcs), a process algebra designed for simulating biological systems (Boemo et al., 2020). In this framework, each component of the system is treated as a process capable of executing certain actions, each governed by an exponential rate. The simulation uses a modified Gillespie algorithm, allowing multiple processes to run in parallel, a property that is especially important for modelling DNA replication, where many events occur concurrently. In our case, we represent replication with three processes: replication origin firing (ORI), and passive replication by left (FL), and right-moving forks (FR). Each process is associated with a specific position at a given resolution (1 kb, in our case) on a chromosome of length L , and origins have an additional parameter, the firing rate, f , or f in our model (Figure 6f).

In each bcs simulation, when a process is activated and its associated action is executed (i.e. when a random variable is realised), the time and location of that event is recorded. Because our model operates at a 1 kb resolution, each action is assigned to a specific 1 kb segment; consequently, the firing of an origin (ORI) or the passive replication by a fork (FL or FR) is registered as an event within that segment. All sites are treated as potential origins; however, the fitting algorithm effectively turns them "on" or "off" by assigning them high or near-zero firing rates, respectively. The model is flexible and can be adapted to different resolutions. In our current implementation, an origin is defined as corresponding to a 1 kb segment. This computational implementation is independent from the analytical approach described above, providing a means to confirm the closed-form mathematical analysis and explore additional replication features. Further details on the bcs formalism and its usage are discussed in SN2.1."

Comment 3.2

In particular, the authors need to justify that their model isn't circular: it start with replication timing data as input, and using those data predicts replication timing for the same samples, which are then compared to the original data. It would seem that any deviation between the two would solely represent the limitations of the model. In particular, late replicating regions are known to be less consistent, more noisy, and more influenced by repetitive sequences, GC content and other technical influences, so it wouldn't be a surprise to find misfits concentrated in late-replicating areas. They also find misfits concentrated near telomeres and centromeres, which again are regions known for relatively poor quality replication timing data due to repetitive sequence and smoothing artifacts. Late replication is correlated with chromosome fragility, overall low gene density but nonetheless an enrichment of long genes, and other properties some of which the authors report as results. So it remains possible that a large fraction of the reported findings are due to model limitations related to data quality in late-replicating regions and to correlations of biological attributes with late replication. Most of the biological insights reported in the paper are not novel, and it is not convincing that the model predicts them as biological rather than technical outcomes. Differently stated, it is not convincing that the model quantifies replication stress and other attributes of replication dynamics.

We thank the reviewer for raising this concern about potential circularity and data-quality artifacts in late-replicating regions. Our model does indeed use replication-timing data both as input and as a benchmark; however, it is not circular. This is because our key assumptions—constant fork speed and stochastic, exponentially distributed origin firing—place strict limits on the steepness of replication-timing transitions. Any instance in which the observed data demand a steeper slope (or slower replication) than our fixed-speed model can accommodate automatically produces a misfit. These misfits therefore reflect deviations from a replication program that proceeds without impediments caused by loci of frequent fork stalling or slowing, rather than merely absorbing them into adjustable parameters. We do concede that certain genomic regions (particularly centromeres and telomeres) are not expected to fit well: they contain highly repetitive sequences and are often show poor mapping quality, thus making timing estimates more prone to noise. In light of these technical issues, we exclude centromeres and telomeres from our main analyses and focus instead on the rest of the genome, where Repli-seq coverage is more reliable.

Crucially, many of our high-misfit regions fall outside centromeres and telomeres and coincide with well-known, experimentally confirmed fragile sites (e.g., FRA16D, FRA3B). These loci are documented hotspots for replication stress and breakage; hence, seeing them emerge naturally from our model's residuals strongly supports a biological rather than purely technical explanation. Moreover, we performed additional statistical checks (Supplementary Note 2.4) confirming the iterative fitting scheme does not systematically inflate firing rates, ensuring the misfits we observe reflect true deviations in replication timing. In this way, our framework serves as a genome-wide screening tool to flag potential replication-stress hotspots (including fragile sites), which researchers can then investigate further using orthogonal experiments. We believe that these findings establish our method as a robust framework for identifying replication stress hotspots that warrant further investigation using independent validation methods (see also Reviewer #1, Comment 1.1).

Comment 3.3

The authors disclose that "Misfits also vary between cell lines, with HCT displaying a distinct pattern likely due to differences in data processing". This would be consistent with technical artifacts influencing at least some of the results.

We thank the reviewer for this insightful comment. Although we do observe some variation in misfit patterns across cell lines, for example, the distinct pattern in HCT, which we attribute in part to differences in data processing, our analyses also reveal consistent trends across all 11 cell lines. In particular, our detailed examination of HCT116 (discussed in the revised section on HCT116 in the Supplementary Information) shows that misfits correlate with known markers of replication stress, supporting the biological relevance of our findings. Our model is designed as a broadly applicable framework for identifying potential replication stress hotspots rather than a tool that is narrowly focused on cell-specific replication programs. We acknowledge that technical artifacts may influence the results in certain cell types, but our approach also underscores the conserved relationships between replication timing, misfits, and genomic features across diverse cell lines.

Comment 3.4

In the discussion, the authors cite Gindin et al, 2014 with regards to incorporating Hi-C data, however Gindin et al did not use Hi-C data in their paper.

We thank the reviewer for pinpointing this, and we apologise for the lapse. We originally cited that reference to highlight a model that incorporates chromatin structure. We have now replaced it with the more pertinent citation, "Control of DNA replication timing in the 3D genome" (Marchal, Sima, and Gilbert, 2019).

Papers cited in this revision

- Zhao, Peiyao A., Takayo Sasaki, and David M. Gilbert. "High-resolution Repli-Seq defines the temporal choreography of initiation, elongation and termination of replication in mammalian cells." *Genome biology* 21 (2020): 1-20.
- Boteva, Lora, et al. "Common fragile sites are characterized by faulty condensin loading after replication stress." *Cell reports* 32.12 (2020).
- Letessier, Anne, et al. "Cell-type-specific replication initiation programs set fragility of the FRA3B fragile site." *Nature* 470.7332 (2011): 120-123.
- Le Tallec, Benoît, et al. "Molecular profiling of common fragile sites in human fibroblasts." *Nature structural & molecular biology* 18.12 (2011): 1421-1423.
- Barlow, Jacqueline H., et al. "Identification of early replicating fragile sites that contribute to genome instability." *Cell* 152.3 (2013): 620-632.
- Pentzold, Constanze, et al. "FANCD2 binding identifies conserved fragile sites at large transcribed genes in avian cells." *Nucleic acids research* 46.3 (2018): 1280-1294.
- Tubbs, Anthony, et al. "Dual roles of poly (dA: dT) tracts in replication initiation and fork collapse." *Cell* 174.5 (2018): 1127-1142.
- Massey, Dashiell J., and Amnon Koren. "High-throughput analysis of single human cells reveals the complex nature of DNA replication timing control." *Nature Communications* 13.1 (2022): 2402.

- Dileep, Vishnu, and David M. Gilbert. "Single-cell replication profiling to measure stochastic variation in mammalian replication timing." *Nature communications* 9.1 (2018): 427.
- Boemo, Michael A., Luca Cardelli, and Conrad A. Nieduszynski. "The Beacon Calculus: A formal method for the flexible and concise modelling of biological systems." *PLoS computational biology* 16.3 (2020): e1007651.
- Marchal, Claire, Jiao Sima, and David M. Gilbert. "Control of DNA replication timing in the 3D genome." *Nature Reviews Molecular Cell Biology* 20.12 (2019): 721-737.

Reviewer #1

Comment

It is unfortunate that it is difficult to readily analyze the ERFs, but the authors have made a reasonable attempt to address the points raised by this reviewer, improving the quality of the paper. Therefore, I would like to support its publication in Nature Communications.

We sincerely thank Reviewer #1 for the supportive comments and for recognizing our efforts. The reviewer's supportive remarks and recommendation for publication in *Nature Communications* are greatly appreciated.

Reviewer #2

Comment

*The authors answered most comments to my satisfaction. However, they did not test their method on *S. cerevisiae* as suggested in my previous report. I believe this is an important "control experiment" of their method: it should be able to recover the location of the most active origins in that organism, which are well known. This would also help with concerns about the circularity of the method, as expressed by one other referee. If that is done, I would recommend the paper for publication.*

We thank the reviewer for emphasizing the value of a *S. cerevisiae* "control experiment". In response, we have implemented the full fitting pipeline on budding-yeast replication-timing data (Müller et al., 2014) and compared the inferred firing-rate peaks with the curated origin database (OriDB; Siow et al., 2012). We found that the model recovers >86 % of the origins at high-firing rates labelled as 'Confirmed' or 'Likely' (as per OriDB nomenclature) within ± 2 kb, and the fitted firing-rate landscape strongly correlates with independent measures of origin location (Mann–Whitney U test and point-biserial correlation, $p < 10^{-12}$). These results are presented in the new Supplementary Note 2.6 ('Theoretical digression: Application to *Saccharomyces cerevisiae*') and demonstrate that the method can accurately retrieve known origin positions without circular bias, thereby addressing the reviewer's control-experiment request and the circularity concerns raised earlier.

Reviewer #3

Comment

I appreciate the authors' efforts in responding to my (and the other reviewers) comments. While the manuscript has been improved, I still find it more appropriate for a more specialized journal; my main concern, that many of the results are due to noise in late-replicating regions and close to chromosome ends/gaps, remains. I have not seen a reproduction of the results when controlling for the late replication timing of fragile sites and long genes for instance, and the biological novelty of the findings remains limited. While the analysis and methods are sophisticated and the paper very well presented, it falls short of providing a sufficient addition to the understanding of DNA replication biology.

We thank the reviewer for highlighting these important points and fully appreciate the concern that technical noise near late-replicating domains and chromosome ends could inflate our results. As noted in the

previous revision, we already excluded all centromeres, telomeres, assembly gaps, and low-mappability bins (see SN 2.3 and 2.4); these excluded regions are shaded in Figure S3 and omitted from every enrichment test. To make this explicit, we have inserted the following clarifying line:

Page 6 (in Supplementary Information; SN2.4.1 *Misfit regions in HCT116*):

“All downstream analyses exclude centromeres, telomeres, assembly gaps, and bins with low mappability.”

Even after this filtering, the correlations with fragile sites remain highly significant (Figure S4), supporting the notion that these results are of biological significance rather than artefacts. In addition, and in response to Reviewer #2, we show that the same closed-form pipeline recovers the majority of confirmed origins in budding yeast (SN 2.6), demonstrating that the approach is neither circular nor restricted to large, repeat-rich genomes, an issue previously raised by the present reviewer. Taken together, we believe these advances, both methodological and in the systematic identification of replication-stress candidates across multiple human cell types, offer broad interest that is well aligned with the readership of *Nature Communications*.

Note on Figure 1 attribution

Reviewer #3 has also raised the issue regarding figure reuse (Figure 1a), which we agree should be properly attributed. This figure was inspired by that of Hulke et al. (2020) and indeed this citation was in an earlier draft but was removed in error. We are therefore very grateful to the reviewer for making us aware of this error and the citation has now been restored in the caption of Figure 1.

Papers cited in this revision

- Müller, Carolin A., et al. "The dynamics of genome replication using deep sequencing." *Nucleic acids research* 42.1 (2014): e3-e3.
- Siow, Cheuk C., et al. "OriDB, the DNA replication origin database updated and extended." *Nucleic acids research* 40.D1 (2012): D682-D686.
- Hulke, Michelle L., Dashiell J. Massey, and Amnon Koren. "Genomic methods for measuring DNA replication dynamics." *Chromosome Research* 28.1 (2020): 49-67.

Reviewer #2

Comment

The authors' responses address my only remaining concern, so I recommend that the paper be accepted for publication.

We would like to thank Reviewer #2 for the positive feedback and the recommendation for publication.

Reviewer #3

Comment

The authors have addressed all of my concerns, thank you.

Likewise, we thank Reviewer #3 for confirming that all concerns have been addressed.